# Continuum Robots: From Conventional to Customized Performance Indicators

**DOI:** 10.3390/biomimetics8020147

**Published:** 2023-04-06

**Authors:** Matteo Russo, Elie Gautreau, Xavier Bonnet, Med Amine Laribi

**Affiliations:** 1Department of Industrial Engineering, University of Rome Tor Vergata, Via del Politecnico 1, 00133 Rome, Italy; 2Faculty of Engineering, University of Nottingham, Jubilee Campus, Nottingham NG8 1BB, UK; 3Department GMSC, Pprime Institute, CNRS—University of Poitiers—ENSMA, UPR 3346 Poitiers, France; 4CEBC Center of Biological Studies of Chizé, CNRS & University of la Rochelle, Villiers—en—Bois, UMR 7372 Deux-Sèvres, France

**Keywords:** continuum robots, robot optimization, robot performance, bioinspired robots, performance indicators

## Abstract

Continuum robots have often been compared with rigid-link designs through conventional performance metrics (e.g., precision and Jacobian-based indicators). However, these metrics were developed to suit rigid-link robots and are tuned to capture specific facets of performance, in which continuum robots do not excel. Furthermore, conventional metrics either fail to capture the key advantages of continuum designs, such as their capability to operate in complex environments thanks to their slender shape and flexibility, or see them as detrimental (e.g., compliance). Previous work has rarely addressed this issue, and never in a systematic way. Therefore, this paper discusses the facets of a continuum robot performance that cannot be characterized by existing indicator and aims at defining a tailored framework of geometrical specifications and kinetostatic indicators. The proposed framework combines the geometric requirements dictated by the target environment and a methodology to obtain bioinspired reference metrics from a biological equivalent of the continuum robot (e.g., a snake, a tentacle, or a trunk). A numerical example is then reported for a swimming snake robot use case.

## 1. Introduction

Continuum robots are characterized by backbones capable of continuous, actuatable bending [1,2,3]. Due to their structural flexibility, these elongated robots can navigate across complex environments, crawling through narrow and tortuous paths where conventional rigid-link robots cannot operate. These features open a wide range of practical solutions when movements and displacements are strongly constrained, notably for industrial maintenance and repair of aeroengines, in nuclear plants, telecommunication ducts [4,5], teleoperated keyhole, or during endoscopic surgery [6,7].

A wide range of designs have been proposed for continuum robots, varying both in backbone architecture and in actuation principle [2]. Successful architectures include robots with a single central backbone [8,9] or a segmented backbone with rigid disks connected by compliant joints (e.g., spring-like [10,11] or twin-pivot [4,12] joints), structures made of sliding disks in series [5], concentric pre-bent superelastic tubes [13,14], and superelastic notched tubes [15]. Actuation methods range from externally driven tendons [4,5,6], linear and rotational motors combined with intrinsic elasticity [13,14], pneumatics [16,17,18], and smart materials [12].

The performances of these design solutions have been evaluated and compared using metrics and frameworks conceived for conventional robotic systems [19,20]. This way, a widespread performance indicator in the continuum robot literature is precision, either in terms of accuracy or repeatability. As recently discussed [21], measuring robot position error provides a mean to evaluate the performances of both hardware and modeling/control methodology, and provides indirectly a measure of stiffness because lower stiffness results in a higher unwanted displacement at the tip. Absolute precision metrics are useful to optimize a specific design [22], for example when implemented into feedback closed loops to control tip position (the objective function to minimize) [23] or to shape the backbone [24,25]. However, they are difficult to implement to compare the performances of continuum robots due to strong disparities of the designs and size of the robots. This issue has been addressed by normalizing the absolute error to the length of the backbone, thereby providing a relative index [26,27]. While this method represents a step forward to standardize precision metrics, it sets as a reference dimension only the length of the backbone, neglecting the influence of other key geometrical variables on precision, notably the outer diameter and bending radius. For example, two robots with the same expected accuracy might exhibit very different behaviors if they have different cross-section diameters.

A different approach to evaluate the performance of continuum robots involves the Jacobian-based metrics, such as manipulability or conditioning number [20]. For example, the dexterity of a three-segment continuum robot can be computed [28,29]. The Jacobian-based metrics are also used to evaluate wrench transmission [30,31]. However, only robots with 6 degrees of freedom have been considered, forsaking one of the key features of the targeted design (i.e., a continuously bending backbone) due to the use of a discrete modeling that resembles that of a conventional manipulator. This suits well the Jacobian-based metrics, as they appropriately characterize the performance of non-redundant manipulators and neglect any further degree of freedom.

Overall, previously used indicators might fail to capture a substantial part of the behavior of the robot. They characterize a small portion of the backbone, limited to the analyzed degrees of freedom. This issue was partially addressed: rather than reducing the Jacobian from a 6 × n (where n is the number of active degrees of freedom of the robot) to a 6 × 6 matrix, an expanded n × n Jacobian was obtained by projecting vectors on the null space of the Jacobian [32]. These vectors fully solve the redundant or hyper-redundant problem by including non-primary tasks or objectives to the computation. However, while providing a working solution to redundancy, this task-oriented procedure does not characterize the specific behavior of the body and backbone of the robot.

These caveats illustrate the more general issues of adopting metrics or performance evaluation methods designed for a fundamentally different type of mechanical system, and they show how conventional performance metrics applied to continuum manipulators can generate critical shortcomings and misrepresentation. 

Another limitation of the use of conventional indicators should be considered. Evaluating continuum robots with indicators designed for rigid-link systems could lead to underestimating the overall capabilities of continuum robots and compromising their future impact: indeed, these metrics are intrinsically biased in favor of rigid-link systems (e.g., a higher stiffness will almost always result in a higher accuracy and repeatability). Deceptively, continuum robots might be considered inferior to conventional manipulators. Nevertheless, their compliance represents their key advantage over rigid-link robots, even if this feature is considered detrimental in most performance metrics. Their ability to configure their backbone along complex curves and paths is not taken into account by any existing index. Finally, conventional performance metrics are mostly adapted to industrial manipulation demands, and do not include the wide array of development and possible utilization of continuum robots.

In this paper, we address these issues by proposing new performance metrics tailored to continuum robots. Two sets of indicators are introduced to describe the geometric features and the kinetostatic behavior of the backbone. The geometry of the robot is defined through a slenderness parameter, which characterizes the robot’s form factor in both reach and cross-section area, and its minimum bending radius, which outlines its motion limits. The kinetostatic behavior is defined through a local stiffness map along the backbone length, allowing a bioinspired optimization of the robot by mimicking the local stiffness of similar biological entities (for example, snakes, tendrils, trunks). The usefulness of these metrics in design optimization is discussed. A numerical example is then reported to illustrate the applicability of the proposed metrics, and the outcomes are used to highlight their advantages and disadvantages. 

## 2. Materials and Methods

### 2.1. Defining Requirements

The narrow, tortuous passages that are often found in the environments where continuum robots can operate are characterized by three different critical parameters: the distance from the access port to the desired workspace, the width of the narrowest passage, and the sharpest bend along the path, as outlined in Figure 1.

These features reflect the design parameters of a continuum robot, shown in Figure 2, as follows:The distance from the entrance or access port * (i.e., the insertion point of the continuum robot) to the desired workspace determines the appropriate length of the backbone of the continuum robot. This dimension must take into account the curvature of the shortest path that the robot can access, plus collision risks and other constraints.The cross-section of the robot must fit the narrowest passage along the navigation path. As such, for a circular cross-section, the external diameter must be smaller than the width of the strictest choke point in the path. In the case of non-circular cross-sections, the diameter of the circle circumscribed to the outer edge can be used instead as a conservative estimation.The geometry of the sharpest bend along the navigation path defines the minimum bending radius that the backbone must achieve.

Another requirement for manipulation is represented by the maximum payload $F_{max}$ that the robot can lift. Although this parameter is not directly related to the geometry of the robot, it affects motion as backbone deflection under load. This relation can be expressed with a stiffness coefficient, which is related to material mechanics (e.g., Young’s modulus) and geometry (e.g., cross-section shape and dimension) through a beam model.

As such, four different specifications were used to characterize a continuum robot: backbone length, cross-sectional diameter, backbone’s bending radius, and payload. These four variables, while useful to evaluate a specific design in a particular use case, suffer from two limitations:*Local parameters.* Whereas backbone length is a fixed parameter, diameter and bending radius are local variables that can have different values at different backbone lengths; payload is similarly evaluated at the tip but does not properly characterize the behavior of different backbone segments.*Absolute metrics.* The proposed specifications are not suited to compare different continuum robot designs. When evaluating the form factor of a continuum robot, we favor a smaller diameter and bending radius, and a longer backbone. Absolute values, however, can complicate comparison of designs at different scales. For example, is a smaller but shorter continuum robot with a 10 mm diameter and bending radius and a 50 mm length better than a larger but longer one with 20 mm diameter and bending radius, but 1000 mm length?

Evidently, global and relative metrics (as defined in [20]) are needed for a meaningful performance evaluation and comparison, and some (such as the slenderness factor proposed in [7]) have been proposed in the literature. In the next sections, we propose global and relative metrics for continuum robots, with systematic definitions that expand previous approaches, and discuss their relationship to local and absolute indicators.

### 2.2. Slenderness

As a key kinematic parameter of the robot, length is generally provided for previous designs in the related literature. The outer cross-sectional diameter is sometimes mentioned, even if many papers on tendon-driven continuum robots only specify the radius at which the tendon are routed, which is representative of the size of the robot, but does not define its encumbrance. The ratio between backbone length $\ell_{b}$ and cross-section outer diameter $d_{0}$ has often been used as a dimensionless parameter that expresses how “thin” or “slender” a continuum robot is (e.g., see the first table in [7]), as slenderness σ:(1)$$\sigma =\frac{\ell_{b}}{d_{0}}$$

In Equation (1), the cross-section outer diameter $d_{0}$ is implied as constant. While this assumption holds true for some designs, others are characterized by a variable diameter. In the second case, the maximum cross-section outer diameter can be used instead for the slenderness ratio to reflect the worst-case behavior. The solution proposed here slightly differs from that in [7], which uses the most proximal diameter (i.e., at the base or fixed end of the continuum robot) instead, assuming a constant or decreasing diameter from proximal to distal (tip) segments. While this is a common solution in variable-diameter continuum robots, using the overall maximum diameter (as illustrated in Figure 2) results in a safer estimation as it also considers the rare continuum robots with a diameter increase at a generic backbone length.

Although the formulation in Equation (1) provides a global design index, a local performance indicator might be more suited to specific cases. An example of this is when a robot navigates in an environment such as that in Figure 1, where the narrowest passage is not at the entrance. If designing the robot with slenderness only in mind, we would be constrained to a diameter $d_{0}$ throughout the whole backbone. However, the proximal segment of the backbone never reaches the choke point, and it could be designed with a larger cross-section diameter. To take this issue into account, a variable cross-section diameter function $d_{0,b}\left(l\right)$ can be mapped along the backbone curve coordinate $l\in \left[0;\;\ell_{b}\right]$ as a local performance indicator.

### 2.3. Flexibility

Neither the two-parameter (backbone length, cross-section diameter) approach nor the slenderness ratio provide any information about how flexible a continuum robot is, which could be defined as the maximum bending angle $\vartheta_{0}$ that can be obtained for a given backbone segment with known length ℓ. A potential parameter to describe this behavior is the bending radius ρ, related to arc length ℓ and angle ϑ through
(2)ℓ=ϑρ

A minimum bending radius can be thus defined for the segment as $\ell /\vartheta_{0}$. This minimum bending radius depends on various factors, including material properties (e.g., maximum elastic strain sustainable by the backbone), design features (e.g., collision between consecutive disks/vertebrae in a tendon-driven continuum robot), and/or actuation or control limits. From a kinematic perspective, all these factors set a lower boundary to ρ, even though from a physical perspective distinct results are possible: if the radius is bounded by material limits, the backbone can physically overcome the boundary but be damaged in the process (for example, by moving past the elastic behavior of the backbone and induce a plastic deformation); a physical constraint (e.g., collision between consecutive disks or notches fully closing) can instead be actively exploited to lock the robot in the limit position for a stiffness increase at a known bending. As such, understanding the physical principle that constrains ρ is critical to avoid failure and potentially improve performance.

Different segments of the backbone can be characterized by different values of minimum bending radius. As continuum robots often move with a follow-the-leader strategy, with body shape following tip trajectory [24], a very flexible tip can easily be hindered by a less-flexible proximal segment. As a low flexibility in any segment of the backbone can thus act as bottleneck for the performance of the whole robot, the flexibility of the whole robot should be computed with the maximum value of minimum bending radius $\rho_{0}$.

While slenderness is a dimensionless quantity, the minimum bending radius is a distance, and thus, it can be expressed in meters or millimeters. For a dimensionless flexibility index φ, the following formulation is proposed:(3)$$\varphi =\frac{\ell_{b}}{\rho_{0}}$$

Similar to slenderness σ, the higher the flexibility φ is, the better the performance of the robot (differently from the minimum bending radius, which is preferable in smaller values). It is worth noting that this formulation includes the length of the entire backbone $\ell_{b}$, which is different from the segment length ℓ considered in Equation (2). However, for a continuum robot bent with continuous curvature throughout its body, Equation (3) can be rewritten for $\ell =\ell_{b}$ as $\varphi =\vartheta_{0}\left(\ell_{b}\right)=\vartheta_{b}$. As such, the flexibility φ also represents the maximum angle of bending (in radians) achievable by the robot when fully coiled on itself.

As discussed for slenderness, Equation (3) is a global index, but a local performance indicator could better suit paths where the sharpest bend is far from the entrance. In this case, a variable minimum bending radius function $\rho_{0,b}\left(l\right)$ can be mapped along the backbone curve coordinate $l\in \left[0;\;\ell_{b}\right]$ as a local performance indicator.

### 2.4. Stiffness

Parameters $\ell_{b}$, $d_{0}$, and $\rho_{0}$, as well as slenderness σ and flexibility φ, characterize the geometry of a continuum robot but cannot represent its behavior under load. In conventional rigid-link robots, statics and dynamics are described through either Jacobian-based indicators, such as dynamic manipulability or stiffness [20], or force transmissibility [33]. However, Jacobian-based indicators only characterize the behavior of a limited segment of the backbone, whereas force transmissibility relies on lumped parameters on rigid joints and cannot represent a compliant backbone. The latter does not affect “classical” snake-like robots that are made of a series of identical actuated rigid modules, for example with servomotors [34] or magnetic actuation [35]. Such robot designs do not consider the cross-section profile variation and compliancy that characterize many continuum (and soft) robots, as for example in the tail of fluid-driven [36] or cable-driven [37,38] tail mechanisms of fish-like robots. To characterize the compliance of a continuum robot backbone, the relationship between load and corresponding motion thus requires a stiffness parameter independent from the Jacobian and can integrate material mechanics, which can be estimated as the ratio between an applied load at the tip F and the corresponding tip deflection δ:(4)$$k=\frac{F}{\delta }$$

The practical computation of this value depends on the mechanical model used to describe the body of the continuum robot. A geometrically exact solution can be obtained by defining the robot body as a beam through Euler–Bernoulli, Kirchhoff, or Cosserat beam theories [39,40]. For each segment with length ℓ of the robot backbone, independently from the robot design, a local stiffness coefficient can be then evaluated through material properties (such as Young’s modulus E) and geometry (such as second moment of area I) when material mechanics are linear or can be linearized. In these cases, the result is also independent from force and deflection. For example, the stiffness coefficient of a cantilever beam with a load at the tip can be computed as $k_{E}=3EI/\ell^{3}$ with Euler–Bernoulli theory. Non-linear properties result in more complex formulations, but they can still be applied to any kind of design by assigning a stiffness function, rather than a value, to each backbone segment.

The formulation in Equation (4) reports a local evaluation of the backbone stiffness. When global metrics are needed, an average stiffness value can be used as reference instead, evaluated as
(5)$$k_{a}=\frac{1}{\ell_{b}}\;\int_{0}^{\ell_{b}}k\left(l\right)dl$$
when a continuous stiffness function k(l) is available, or as
(6)$$k_{a}=\frac{1}{\ell_{b}}\;\overset{n}{\underset{i=1}{\sum }}\ell_{i}k_{i}$$
in the case of discrete formulations (continuum robot with *n* segments; each *i*th segment of length $\ell_{i}$ has homogeneous stiffness $k_{i}$). 

### 2.5. Remarks

In this section, we have discussed the performance of continuum robots, from the operational requirements to the parameters that are influenced by them. As a result, we proposed two global performance metrics, summarized as follows:

**Definition** **1.**
*Slenderness: the ratio between the length of the backbone of the robot, measured as the distance between its extremities along the centerline of the robot body, and the maximum cross-section diameter, measured as the largest diameter of a circle that circumscribes any cross-section of the robot body.*


**Definition** **2a.**
*Flexibility (radius of curvature): the ratio between the length of the backbone of the robot, measured as the distance from its most proximal to its most distal point along the centerline of the robot body, and the minimum bending radius of the backbone, measured as the minimum radius of curvature that can be achieved by the backbone when bent with a constant curvature.*


**Definition** **2b.**
*Flexibility (bending angle): the maximum angle that can be subtended by the backbone centerline curve when the backbone is bent with a constant curvature.*


Further local metrics have been proposed to characterize the varying behavior of a continuum robot along its backbone. As the backbone length is a fixed value, a local evaluation of both slenderness and flexibility can be obtained with local values of maximum cross-section diameter and minimum bending radius, which might differ from the global maximum/minimum. The dynamic behavior of a segment of the robot can be characterized with a local stiffness value, which represents the ratio between load and deflection. All these indicators and parameters are summarized in Table 1.

As noted in Table 1, the geometrical requirements bound the value of the geometrical parameters of the robot. However, once those requirements are met, a further “improvement” of the design (that is, by reducing $d_{0}$ and $\rho_{0}$) increases slenderness and flexibility without affecting the performance of the robot in the desired environment: the robot is either able to navigate through it or not. Further, a reduction in $d_{0}$ and $\rho_{0}$ open technological or manufacturing perspectives, as the lower bound of these values is set by manufacturing tolerances, material strength, and empty cross-section volume required to fit motors, tools, sensors, and wires for the end-effector. Conversely, matching the stiffness to a desired value (as measured on the biological equivalent) does not intrinsically enable or disable specific tasks, but only changes ”how well” the robot performs.

To obtain an optimal design in terms of the proposed performance metrics, a single-objective optimization algorithm can be used, which minimizes the difference between the desired and actual stiffness of the robot as
(7)$$\min\left|k-k_{\mathrm{des}}\right|$$
subject to the geometrical requirements as constraints:(8)$$\ell_{b}>\ell_{\min};d_{0}<d_{\max};\rho_{0}<\rho_{\max}.\;\;$$

This formulation enables an efficient optimization for a task-oriented design. Notably, depending on the material and mechanics models adopted for the optimization, the stiffness coefficient can be written as a function of a geometrical parameter, further simplifying the optimization problem. For example, a Euler beam model such as that in Equation (7) can tune stiffness by varying cross-sectional diameter only, once all the other parameters (Young’s modulus, cross-section shape) are fixed. 

On the other hand, when a design is developed to be generic rather than fitting specific task requirements, a better result could be obtained with a multi-objective optimization with slenderness, flexibility, and stiffness (encompassing material selection and cross-section shape and size) as objective functions.

## 3. Results and Discussion

This section reports two study cases to illustrate how the proposed performance metrics can be used as a tool for optimal design (with a bioinspired numerical study), and as reference for robot comparison (by reviewing selected continuum robot designs).

### 3.1. Performance Metrics as Design Tools: A Bioinspired Example

A numerical example of design optimization using the proposed framework is reported. As discussed above, the geometrical requirements of a continuum robot can be extracted from the operational environment, which sets clear boundaries and constraints. However, defining a target value $k_{\mathrm{des}}$ or the distribution range for this stiffness parameter is a challenging task: a non-optimal stiffness is difficult to detect as it can reduce efficiency without precluding functioning. As a strict boundary to stiffness values can hardly be set a priori, a bioinspired solution is proposed.

Continuum designs have always been inspired by nature: mobile continuum robots were developed to mimic the locomotor versatility of snakes that can explore galleries and crevices [41], continuum manipulators were conceived after tentacles [42] and elephant trunks [43]. Such biological examples, honed by millennia of evolution, can be studied to extract a target stiffness value or distribution for continuum robots. Different continuum robot applications, such as swimming, slithering, or manipulation, refer to different biological counterparts. In this section, we propose an example procedure to extract a reference stiffness coefficient to improve the swimming performances of a bioinspired robot, although the same methodology can be extended to other applications.

The example focuses on a snake-like continuum robot that intends to mimics snake undulations and more generally the main locomotion mode of anguilliform swimmers (e.g., eels). This specific case involves statics, as many continuum manipulation tasks [44,45], and dynamics, with time-varying external forces induced by the water flow. Indeed, a dynamic model is required to predict the shape of the robot while swimming [46].

To define a target stiffness distribution for this swimming snake-like robot, studies have been performed on both snake locomotion and body mechanical properties. A video analysis software was developed to acquire swimming snake cone cones (i.e., image staking to visualize the relative position of each body part during undulatory cycles), head amplitude, head frequency, and head velocity in the function of time [47]. Presumably, these data represent an optimized biomechanical solution that can be used as a standard with which the snake robot can be designed and actuated, acting as reference values for the geometric parameters previously defined. The snake vertebrae length and body profile were measured on two snake species (*Hierophis viridiflavus*, *Natrix helvetica*) [48] and used to size the snake robot vertebrae length and disk diameter. This provided the core data to shape the broad snake geometry.

Then body stiffness was measured on different segments of dead snakes in the four main bending directions (lateral side left and right, dorsal, and ventral). The deflection angle was measured in function of the force applied at the tip of selected segments of the snake, which results in an equivalent bending moment allowing us to size the compliant vertebrae of the robot. The maximum angle between two vertebrae along the whole snake body, measured using X-rays, allowed us to identify the mechanical stops and to define the maximum bending deflection of compliant vertebrae. Dead snakes were collected opportunistically, for example hit by vehicles (permit issued to XB, DBEC 004/2022). The measured data resulted in respective measurement intervals in which the snake robot skeleton was sized and designed [48], represented through elliptical vertebrae in a 3D plot (Figure 3).

An equivalent beam model with a discrete number of elements (30 cylindrical segments, 40 mm length each, total backbone length 1200 mm) depicts the robot mechanical properties derived from the biological snake body deflections that were empirically measured [48] and numerically characterized as shown in Figure 3. The equivalent beam section parameters that model each vertebra are identified with the Euler–Bernoulli beam theory of a large deflection non-linear elastic beam [49], expressed as
(9)$$EI\;\delta^{\prime \prime }+F\sin\left(\delta \right)=0,$$
where I refers to second moment of area (in our case, for the circular section of an equivalent beam). The equivalent beam diameters $d_{0,b}\left(l\right)$ at different backbone lengths $l\in \left[0;\;\ell_{b}\right]$ can be then estimated by computing an angle δ corresponding to the beam deflection obtained from the empirical measurements.

The optimization resulted in a backbone stiffness distribution that mimics biological snakes, rather than being homogeneous along the backbone (as usually seen in continuum robot designs). The variable deflection resulting from this distribution is reported in Figure 4, where the deformation of the equivalent backbone with circular cross-section is computed through a Finite Element Analysis (FEA). The backbone was simulated with Nylon PA12 (Young’s modulus of 1471 MPa, mechanically tested and characterized by a linear elastic behavior up to 0.016 strain). A fixture was added at the mid-length of the backbone, and a load of 0.1 N was applied to both extremities, resulting in a maximum strain equal to 0.004 (ensuring linear elastic behavior).

The resulting local stiffness is reported in Appendix A (Table A1), for an overall average stiffness $k_{a}$ of 0.88 N/mm. The stiffness value here obtained is a function of Young’s modulus and moment of area, as it has been computed through Euler beam theory. While a cylindrical design is here reported as example, stiffness can be used as a tool to design equivalent robots with different shapes and material, as long as they result in the same value of stiffness coefficient k. By varying these parameter, tighter constraints on flexibility and slenderness can also be met.

### 3.2. Performance Metrics for Design Evaluation: Comparing Existing Continuum Robots

The proposed framework is applied to the optimal design of a novel continuum robot in the previous section, but one of its key applications enables the comparison between different designs at different scales: when comparing two designs of unequal size, both slenderness and flexibility remove the bias of an absolute geometrical value. As such, a review of exemplary existing designs is reported in Table 2 to show how the proposed framework can facilitate performance evaluation.

As shown in the table, the only parameter which is always reported is the backbone length. Outer diameter is reported for almost all the designs, while bending radius is often provided, either explicitly or implicitly (through maximum bending angle per section). Payload information is reported only for specific designs in which it is critical for the proposed operation, while stiffness is almost never discussed.

The proposed slenderness and flexibility metrics highlight trends according to robot architecture (here classified by type of actuation). Pneumatically actuated robots present the lowest performance in form factor but are generally better from a payload and stiffness perspective. Tendon-driven and push-pull designs can achieve good slenderness and the best flexibility, while concentric tube report the best slenderness (outside of outlier [5], whose values are affected by a long passive section) but can be limited in flexibility due to their particular design (based on pre-curvature).

Although Table 2 highlights the advantage of unitless global metrics, enabling the comparison between different continuum robot architectures at different scales, it also shows one of the limitations of general metrics: slenderness and flexibility might not fully capture the behavior of specific designs (e.g., concentric tube robots).

Another challenge in applying the proposed framework to existing robots lies in the exact definition of performance metrics. Noticeably, the values for some of the robots discussed in Table 2 are computed differently in [7]: the values in Table 2 focus on the continuum section of the robot, while in [7], further components (e.g., passive and/or rigid links, support or delivery systems) are considered, thus resulting in distinct values of diameter, length, and slenderness. This issue stems from the lack of an exact definition for what constitutes the backbone of a continuum robot, especially for hybrid or complex designs, as well as from limited documentation (e.g., when a paper only provides the overall length including the end-effector, or when the design is not described in detail).

## 4. Conclusions

Performance indicators are essential tools for the optimal design and selection of robotic systems. However, conventional metrics cannot fully characterize the behavior of continuum robots. Therefore, this paper proposed an analysis and discussion of the key features of continuum robots, identifying key specifications that better describe these systems with a focus on three key metrics:Slenderness, defined as the ratio between the length of the backbone of the robot and the maximum cross-section diameter.Flexibility, defined as the ratio between the length of the backbone of the robot and the minimum bending radius of the backbone.Stiffness, defined as the ratio between an applied load at the tip of a backbone segment and the corresponding deflection.

The main contributions of this work can be summarized in the following points:The design requirements for continuum robots were discussed with respect to their operational environment and task.Tailored metrics for continuum robots were defined to describe their shape factor (slenderness), motion range (flexibility), and dynamic behavior (stiffness).A numerical example of bioinspired design has been reported as an example of the proposed metrics as design tools.A review of a wide variety of designs further extends the usefulness of the proposed metrics to compare different systems at different scales.

The proposed metrics can generally be applied to any continuum system, either for comparison between different designs or as objective functions for design optimization, as shown in the reported examples. As it stands, equivalent indicators have been used in the past for these aims, but their use has been fragmented and past definitions were either not general or limited to specific designs. Thus, the lack of a formal framework hindered the related research.

The proposed flexibility and slenderness, as they refer to the geometry of the robot only, can universally be applied to any continuum robot architecture, even if specific designs (e.g., inflatable robots) push the boundaries of the proposed definitions and could be characterized with multiple values for those indicators rather than a single one (e.g., referring to inflated and deflated body). Stiffness, while general in its definition, can be more challenging to adopt as its formulation depends on the mechanics and material models. An update to it might be required in future as new, more accurate or efficient modeling techniques emerge. 

In the future, we can foresee an expansion to continuum robots of other in-depth metrics that refer to specific facets of performance only (e.g., related to dexterity, conditioning, and force transmission) to integrate and improve the proposed framework.

## Figures and Tables

**Figure 1 biomimetics-08-00147-f001:**
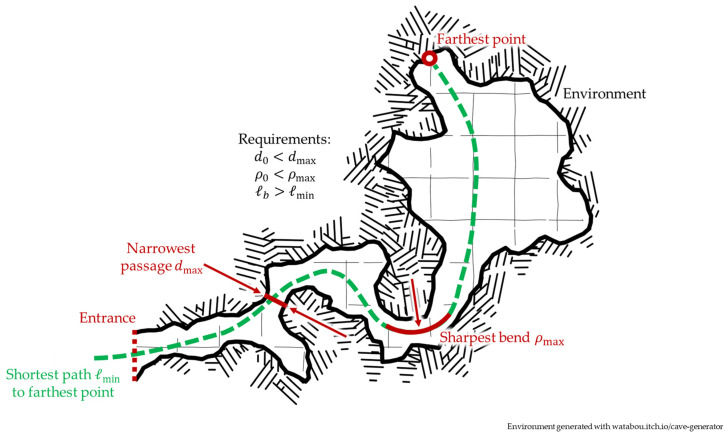
Extracting geometric requirements for continuum robot design from a target environment: the backbone length $\ell_{b}$ must be long enough to reach each point of the workspace along a feasible path (dotted green line in the figure, with curve length $\ell_{\min}$ describing the longest feasible path); the largest cross-sectional diameter $d_{0}$ must fit the narrowest passage (in red in the figure, characterized by choke point width $d_{\max}$); and the backbone’s minimum bending radius $\rho_{0}$ must be smaller than that of the sharpest bend along all feasible paths $\rho_{\max}$ (in red in the figure).

**Figure 2 biomimetics-08-00147-f002:**
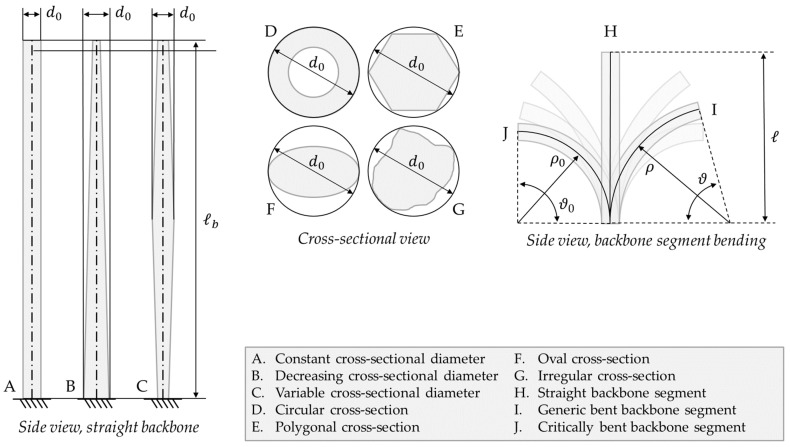
Geometric parameters for continuum robot design: backbone length $\ell_{b}$, measured from base to tip along the centerline; cross-sectional diameter $d_{0}$, defined as the maximum diameter of the circle that circumscribes a cross-section border; and minimum bending radius $\rho_{0}$, which is the smallest curvature radius that a backbone segment can achieve when bending along a circle arc (please note that the critical bending can refer to any angle, not only the 90° represented in the example figure).

**Figure 3 biomimetics-08-00147-f003:**
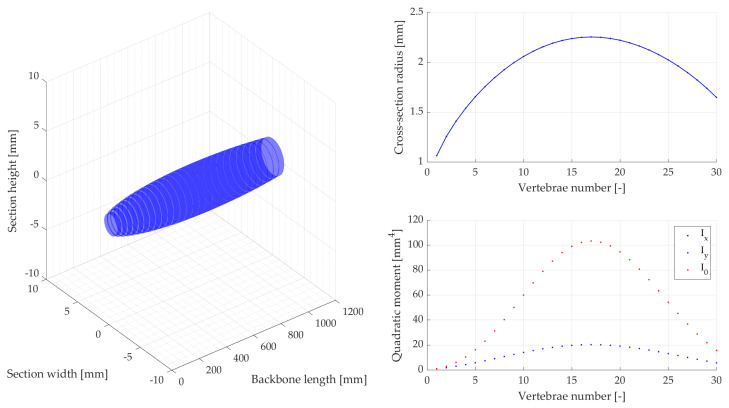
Characterizing backbone behavior through beam elements: 3D representation of the biological data measured on snakes (1200 mm backbone length, 30 vertebrae), with moment of area and cross-section diameter of an equivalent circular cross-section.

**Figure 4 biomimetics-08-00147-f004:**
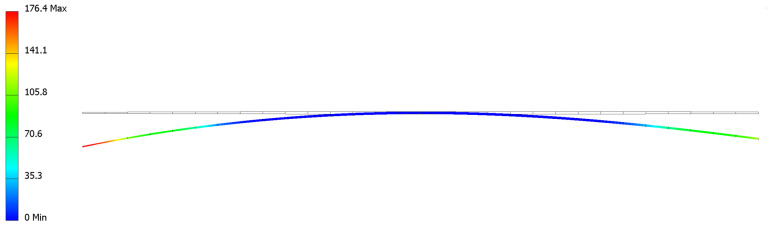
A numerical example of the deformation of a backbone with variable curvature (colormap showing backbone displacement in mm), tested on the equivalent circular beam geometry reported in Figure 3.

**Table 1 biomimetics-08-00147-t001:** Parameters and performance metrics.

Parameter	Symbol	Formulation	Description	Type
Average stiffness	$k_{a}$	Equations (5) or (6)	Average stiffness of the backbone	Index (global)
Backbone coordinate	l	$l\in \left[0;\;\ell_{b}\right]$	Backbone curvilinear coordinate	Design
Backbone length	$\ell_{b}$	-	Max length of robot centerline (base to tip)	Design
Bending angle	ϑ	*-*	Bending angle of a segment ℓ	Variable
Bending radius	ρ	ϑρ=ℓ	Bending radius of a segment ℓ	Variable
Cross-section diameter	$d_{0,b}\;\left(l\right)$	-	Cross-section width at length l	Design
Deflection	δ	-	Deflection caused by external force F	Variable
Desired stiffness	$k_{\mathrm{des}}$	-	Stiffness of the biological equivalent	Requirement
External force	F	-	Load on a generic segment ℓ	Variable
Flexibility	φ	$\varphi =\ell_{b}/\rho_{0}$	Ratio of backbone length and minimum bending radius	Index (global)
Flexibility	$\vartheta_{b}$	$\vartheta_{b}=\ell_{b}/\rho_{0}$	Max angle subtended by coiled backbone	Index (global)
Local min bending radius	$\rho_{0,b}\;\left(l\right)$	-	Minimum bending radius at length l	Design
Max. cross-section diameter	$d_{0}$	$d_{0}=\max\left(d_{0,b}\left(l\right)\right)$	Largest cross-section width of the robot	Design
Maximum bending angle	$\vartheta_{0}$	*-*	Maximum bending angle of a segment ℓ	Design
Maximum payload	$F_{\max}$	-	Maximum payload at the end-effector	Index (global)
Minimum bending radius	$\rho_{0}$	$\rho_{0}=\max\left(\rho_{0,b}\left(l\right)\right)$	Minimum bending radius of the robot	Design
Narrowest passage width	$d_{\max}$	$d_{\max}>d_{0}$	Width of choke point along the path	Requirement
Required reach	$\ell_{\min}$	$\ell_{\min}<\ell_{b}$	Length of longest desired path	Requirement
Segment length	ℓ	-	Length of a generic backbone segment	Design
Sharpest bending radius	$\rho_{\max}$	$\rho_{\max}>\rho_{0}$	Smallest bending radius along the path	Requirement
Slenderness	σ	$\sigma =\ell_{b}/d_{0}$	Ratio of backbone length and maximum cross-section diameter	Index (global)
Stiffness	k	k=F/δ	Stiffness of a segment ℓ	Index (local)

**Table 2 biomimetics-08-00147-t002:** Performance evaluation and comparison of existing continuum robots.

Robot	Ref.	Type	$\ell_{b}$ [mm]	$d_{0}$ [mm]	$\rho_{0}$ [mm]	k [N/m]	$F_{\max}$ [N]	σ [-]	φ [-]
STIFF-FLOP	[50]	Pneumatic	100	32	36	-	47.1	3	2.8
TEPM	[51]	Pneumatic	200	-	176	290	-	-	1.13
Pneumatic robot	[52]	Pneumatic	380	150	200	1.1	50	2.5	1.9
FLARE	[4]	Tendon-driven	715	12	55	11.2	-	60	13
COBRA	[5]	Tendon-driven	5500	9	60	-	-	610	91.7
Extensible robot	[9]	Tendon-driven	165	7	7	-	-	24	24
RAIN	[27]	Tendon-driven	1015	20	-	14.7	0.2	50	-
Medrobotics Flex	[53]	Tendon-driven	170	28	113	-	-	9	1.5
I^2^ Snake Robot	[54]	Tendon-driven	366	16	-	-	2	23	-
HARP	[55]	Tendon-driven	300	12	75	-	5	25	4.0
IREP	[56]	Push-pull	60	6.4	19	-	2	30	3.1
MIS Robot	[57]	Push-pull	37	4.2	12	-	1	8.8	3.1
Surgical Robot	[58]	Concentric tube	420	2.3	-	-	-	182	-
Robotic Catheter	[59]	Concentric tube	830	4.5	-	-	-	184	-

## Data Availability

The data presented in this study are available in this article.

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
