# Peer review of "Continuum Robots: From Conventional to Customized Performance Indicators"

_biomimetics, 2023, doi:10.3390/biomimetics8020147_

Round 1

Reviewer 1 Report

The paper defines a tailored framework of geometrical specifications and kinetostatic indicators for a continuum robot.

The idea is interesting and original. In my opinion, the paper can be accepted for publication: I would like just to add a minor remark concerning the number of indicators which, in my opinion, is quite high. I would suggest to highlight which of the indicators are of uttermost importance, with respect to the others.

Author Response

Manuscript title: “Continuum robots: from conventional to customized performance indicators”

Manuscript ID: biomimetics-2291592

We would like to thank the reviewers for the insightful comments. Please find enclosed our replies to their comments, with the changes highlighted in the revised manuscript.

REVIEWER 1:

Comment 1.1: The paper defines a tailored framework of geometrical specifications and kinetostatic indicators for a continuum robot. The idea is interesting and original. In my opinion, the paper can be accepted for publication: I would like just to add a minor remark concerning the number of indicators which, in my opinion, is quite high. I would suggest to highlight which of the indicators are of uttermost importance, with respect to the others.

Reply 1.1: We have highlighted the three key indicators in the conclusions, with the following revision:

Therefore, this paper proposed an analysis and discussion of the key features of continuum robots, identifying key specifications that better describe these systems with a focus on three key metrics:

  • Slenderness, defined as the ratio between the length of the backbone of the robot and the maximum cross-section diameter.
  • Flexibility, defined as the ratio between the length of the backbone of the robot and the minimum bending radius of the backbone.
  • Stiffness, defined as the ratio between an applied load at the tip of a backbone segment and the corresponding deflection.

Reviewer 2 Report

This paper presents new performance metrics for the design and performance evaluation of continuum robots. As the ones for measuring the geometric features of the continuum robots, slenderness and flexibility that are both dimensionless values were proposed. As the one for kinetostatic features, stiffness was presented. A optimal design utilizing these indices was formulated. A case study was presented to show the effectiveness of the proposal. Though the presented work seems original, the following issues should be addressed.

[Major comments]

1.The proposed "metrics" basically relate to the features of the backbone of a continuum robot.  As the authors mentioned in conclusions, the more in-depth metrics should be considered in the design of a continuum robot. The role of the metrics proposed in this work in the whole design process of a continuum robot should be clarified first, and it is necessary to discuss how effective/useful the proposed metrics are.

2.In the present work, stiffness of continuum robot is defined as the ratio between an applied load at the tip and the corresponding tip deflection. In the wide application ranges of continuum robots, this definition seems to be limited. (According to this definition, continuum robots to which the proposed method can be  applied are limited.) By listing up the application of the continuum robots which this work intends to cover, the reason why the stiffness is defined in this way should be discussed.

3. The authors emphasized that the proposed framework can be applied to the comparison between different designs at different scales. But, the authors did not explain why and in what situations such comparisons are needed.

[Minor comments]

1. In the text, some "continuous" are wrong and seems to be better replaced by "constant".

2. The order of parameters shown in Table 1 should be properly arranged according to some rules (alphabetical, etc).

Author Response

Manuscript title: “Continuum robots: from conventional to customized performance indicators”

Manuscript ID: biomimetics-2291592

We would like to thank the reviewers for the insightful comments. Please find enclosed our replies to their comments, with the changes highlighted in the revised manuscript.

REVIEWER 2:

Comment 2.1: The proposed "metrics" basically relate to the features of the backbone of a continuum robot. As the authors mentioned in conclusions, the more in-depth metrics should be considered in the design of a continuum robot. The role of the metrics proposed in this work in the whole design process of a continuum robot should be clarified first, and it is necessary to discuss how effective/useful the proposed metrics are.

Reply 2.1: The conclusions have been expanded to provide further information of the use (comparison/optimization) and effectiveness (as generality/universality) of the proposed metrics with the following discussion:

The proposed metrics can generally be applied to any continuum system, either for comparison between different designs or as objective functions for design optimization, as shown in the reported examples. As it stands, equivalent indicators have been used in the past for these aims, but they use has been fragmented and past definitions were either not general or limited to specific designs. Thus, the lack of a formal framework hindered related research.

The proposed flexibility and slenderness, as they refer to the geometry of the robot only, can universally be applied to any continuum robot architecture, even if specific designs (e.g., inflatable robots) push the boundaries of the proposed definitions and could be characterized with multiple values for those indicators rather than a single one (e.g., referring to inflated and deflated body). Stiffness, while general in its definition, can be more challenging to adopt as its formulation depends on the mechanics and material models. An update to it might be required in future as new, more accurate or efficient modeling techniques emerge.

In the future, we can foresee an expansion to continuum robots of other in-depth metrics that refer to specific facets of performance only (e.g., related to dexterity, conditioning, and force transmission) to integrate and improve the proposed framework.

Comment 2.2: In the present work, stiffness of continuum robot is defined as the ratio between an applied load at the tip and the corresponding tip deflection. In the wide application ranges of continuum robots, this definition seems to be limited. (According to this definition, continuum robots to which the proposed method can be applied are limited.) By listing up the application of the continuum robots which this work intends to cover, the reason why the stiffness is defined in this way should be discussed.

Reply 2.2: We mention the limitation of the proposed stiffness model highlighted by the reviewer in the revised conclusions, as:

Stiffness, while general in its definition, can be more challenging to adopt as its formulation depends on the mechanics and material models. An update to it might be required in future as new, more accurate or efficient modeling techniques emerge.

We have further revised the paper to discuss the generality of this definition, highlighting how it can theoretically be applied to any kind of continuum robot and is only limited by the modeling techniques used to analyze the robot, rather than its design.

The practical computation of this value depends on the mechanical model used to describe the body of the continuum robot. A geometrically exact solution can be obtained by defining the robot body as a beam through Euler-Bernoulli, Kirchhoff, or Cosserat beam theories [39, 40]. For each segment with length l of the robot backbone, independently from the robot design, a local stiffness coefficient can be then evaluated through material properties (such as Young’s modulus E) and geometry (such as second moment of area I) when material mechanics are linear or can be linearized. In these cases, the result is also independent from force and deflection. For example, the stiffness coefficient of a cantilever beam with a load at the tip can be computed as k_E=3EI/l^3 with Euler-Bernoulli theory. Non-linear properties result in more complex formulations, but they can still be applied to any kind of design by assigning a stiffness function, rather than a value, to each backbone segment.

Comment 2.3: The authors emphasized that the proposed framework can be applied to the comparison between different designs at different scales. But, the authors did not explain why and in what situations such comparisons are needed.

Reply 2.3: The main advantage is given by using relative, rather than absolute, metrics.

To provide an example to clarify, we can compare the designs of two continuum robots: the first has a length of 10m, a diameter of 0.1m, and a bending radius of 1m; the second is 100mm long, with a diameter of 10mm and a bending radius of 10mm. If comparing with absolute numbers, the first design is better in length, worse in diameter, and worse at bending; however, the comparison would be unfair, as it’s not evaluating the form factor of the design but just the scale at which it has been prototype/tested. With the proposed metrics, the slenderness of the proposed robots is 100 and 10, respectively, and their flexibility is 10, thus showing clearly how the first design is superior to the second.

This concept has been specified in a revision to section 3.2, as:

The proposed framework is applied to the optimal design of a novel continuum robot in the previous section, but one of its key applications is enabling the comparison between different designs at different scales: when comparing two designs of unequal size, both slenderness and flexibility remove the bias of an absolute geometrical value.

It is further explained in the paper after Figure 2, as:

  • Absolute metrics. The proposed specifications are not suited to compare different continuum robot designs. When evaluating the form factor of a continuum robot, we favor a smaller diameter and bending radius, and a longer backbone. Absolute values, however, can complicate comparison of designs at different scales. For example, is a smaller but shorter continuum robot with a 10 mm diameter and bending radius and a 50 mm length better than a larger but longer one with 20 mm diameter and bending radius, but 1000 mm length?

Evidently, global and relative metrics (as defined in [20]) are needed for a meaningful performance evaluation and comparison, and some (such as the slenderness factor proposed in [7]) have been proposed in the literature. In the next sections, we propose global and relative metrics for continuum robots, with systematic definitions that expand previous approaches, and discuss their relationship to local and absolute indicators.

Comment 2.4: In the text, some "continuous" are wrong and seems to be better replaced by "constant". The order of parameters shown in Table 1 should be properly arranged according to some rules (alphabetical, etc).

Reply 2.4: We have reviewed the usage of continuous throughout the paper and maintained it when needed (i.e., when referring to a continuously differentiable curve rather than one with constant curvature). Table 1 has been reorganized to be in alphabetical order as suggested.

Reviewer 3 Report

I would like to congratulate the authors for the work done, and the presentation of that in the paper. I think also, that with those results they are filling a void. 

This paper discusses the facets of a continuum robot performance that cannot be characterized by existing indicator and aims at defining a tailored framework of geometrical specifications and kinetostatic indicators. The proposed framework combines the geometric requirements dictated by the target environment and a methodology to obtain bioinspired reference metrics from a biological equivalent of the continuum robot.

I'm already excited how to use the defined parameters in the design process of the continuum robots.

Despite, please considers some observations:

- In Figure 2, case J graphically assumes that the minimum bending radius must be at ϑ0=90deg, which I think, is not the case.

- Line 267 – The quote “measured as the largest diameter of a circle that circumscribes the cross-section of the robot body”, should sound like “measured as the largest diameter of a circle that circumscribes any cross-section of the robot body”, because there could be several circles (with different diameters) about different cross-section of the same robot body.

- Line 275 – There are two font size used in the paragraph.

- Line 294 – At the beginning of the sentence, should be specified: optimal design regarding what? (I assume, regarding robot stiffness).

- Line 372 – For consistency I recommend to use 1200mm (as in figure), not 1.2m.

References

 -The [23] could be replaced with another reference for absolute precision metrics as feedback for closed-loop control, to avoid excessive self-citation.

Author Response

Manuscript title: “Continuum robots: from conventional to customized performance indicators”

Manuscript ID: biomimetics-2291592

We would like to thank the reviewers for the insightful comments. Please find enclosed our replies to their comments, with the changes highlighted in the revised manuscript.

REVIEWER 3:

Comment 3.1: In Figure 2, case J graphically assumes that the minimum bending radius must be at ϑ0=90deg, which I think, is not the case.

Reply 3.1: This point is now clarified in the revised caption as:

Please note that the critical bending can refer to any angle, not only the 90deg represented in the example figure.

Comment 3.2:  Line 267 – The quote “measured as the largest diameter of a circle that circumscribes the cross-section of the robot body”, should sound like “measured as the largest diameter of a circle that circumscribes any cross-section of the robot body”, because there could be several circles (with different diameters) about different cross-section of the same robot body.

Reply 3.2: Thank you for the correction, the sentence has been revised as suggested.

Comment 3.3:  Line 275 – There are two font size used in the paragraph.

Reply 3.3: The formatting error has been fixed.

Comment 3.4:  Line 294 – At the beginning of the sentence, should be specified: optimal design regarding what? (I assume, regarding robot stiffness).

Reply 3.4: We there refer to an optimal design in terms of the proposed metrics, both stiffness (as objective function) and slenderness & flexibility (as constraints). The sentence has been revised for clarity as:

To obtain an optimal design in terms of the proposed performance metrics, …

Comment 3.5:  Line 372 – For consistency I recommend to use 1200mm (as in figure), not 1.2m.

Reply 3.5: The dimension has been fixed for consistency, as suggested.

Comment 3.6:  The [23] could be replaced with another reference for absolute precision metrics as feedback for closed-loop control, to avoid excessive self-citation.

Reply 3.6: We have replaced the reference with:

  1. Campisano, F., Caló, S., Remirez, A. A., Chandler, J. H., Obstein, K. L., Webster III, R. J., & Valdastri, P. (2021). Closed-loop control of soft continuum manipulators under tip follower actuation. The International journal of robotics research, 40(6-7), 923-938.

Round 2

Reviewer 2 Report

Thank you for the detailed responses to the comments. The reviewer confirmed that the comments have been addressed and the manuscript has been properly revised.